# Prediction of Cognitive Decline in Parkinson’s Disease Using Clinical and DAT SPECT Imaging Features, and Hybrid Machine Learning Systems

**DOI:** 10.3390/diagnostics13101691

**Published:** 2023-05-10

**Authors:** Mahdi Hosseinzadeh, Arman Gorji, Ali Fathi Jouzdani, Seyed Masoud Rezaeijo, Arman Rahmim, Mohammad R. Salmanpour

**Affiliations:** 1Technological Virtual Collaboration (TECVICO Corp.), Vancouver, BC V5E 3J7, Canada; h.mahdi@modares.ac.ir; 2Department of Electrical & Computer Engineering, University of Tarbiat Modares, Tehran 14115111, Iran; 3Neuroscience and Artificial Intelligence Research Group (NAIRG), Student Research Committee, Hamadan University of Medical Sciences, Hamadan 6517838736, Iran; 4Department of Medical Physics, Faculty of Medicine, Ahvaz Jundishapur University of Medical Sciences, Ahvaz 6135715794, Iran; 5Department of Integrative Oncology, BC Cancer Research Institute, Vancouver, BC V5Z 1L3, Canada; 6Departments of Radiology and Physics, University of British Columbia, Vancouver, BC V6T 1Z4, Canada

**Keywords:** Montreal Cognitive Assessment, deep learning, hybrid machine learning systems, radiomics, Parkinson’s disease

## Abstract

Background: We aimed to predict Montreal Cognitive Assessment (MoCA) scores in Parkinson’s disease patients at year 4 using handcrafted radiomics (RF), deep (DF), and clinical (CF) features at year 0 (baseline) applied to hybrid machine learning systems (HMLSs). Methods: 297 patients were selected from the Parkinson’s Progressive Marker Initiative (PPMI) database. The standardized SERA radiomics software and a 3D encoder were employed to extract RFs and DFs from single-photon emission computed tomography (DAT-SPECT) images, respectively. The patients with MoCA scores over 26 were indicated as normal; otherwise, scores under 26 were indicated as abnormal. Moreover, we applied different combinations of feature sets to HMLSs, including the Analysis of Variance (ANOVA) feature selection, which was linked with eight classifiers, including Multi-Layer Perceptron (MLP), K-Neighbors Classifier (KNN), Extra Trees Classifier (ETC), and others. We employed 80% of the patients to select the best model in a 5-fold cross-validation process, and the remaining 20% were employed for hold-out testing. Results: For the sole usage of RFs and DFs, ANOVA and MLP resulted in averaged accuracies of 59 ± 3% and 65 ± 4% for 5-fold cross-validation, respectively, with hold-out testing accuracies of 59 ± 1% and 56 ± 2%, respectively. For sole CFs, a higher performance of 77 ± 8% for 5-fold cross-validation and a hold-out testing performance of 82 + 2% were obtained from ANOVA and ETC. RF+DF obtained a performance of 64 ± 7%, with a hold-out testing performance of 59 ± 2% through ANOVA and XGBC. Usage of CF+RF, CF+DF, and RF+DF+CF enabled the highest averaged accuracies of 78 ± 7%, 78 ± 9%, and 76 ± 8% for 5-fold cross-validation, and hold-out testing accuracies of 81 ± 2%, 82 ± 2%, and 83 ± 4%, respectively. Conclusions: We demonstrated that CFs vitally contribute to predictive performance, and combining them with appropriate imaging features and HMLSs can result in the best prediction performance.

## 1. Introduction

Parkinson’s disease (PD) is a progressive neurological condition characterized by the death of dopaminergic neurons in the substantia nigra [1,2]. A recent study [3] indicated a doubled increase in PD from 1990 to 2015. First, PD was introduced as a motor dysfunctionality characterized by bradykinesia, resting tremors, rigidity, and postural instability [4], whilst it was later shown as a complex disease with neuropsychiatric and other non-motor symptoms such as sleep issues, depression, sleep disorders, smell problems, memory issues, attention issues, difficulty with visual-spatial abilities, etc. [5,6]. A common non-motor symptom of PD is cognitive impairment, which ranges from mild to severe [1]. The Montreal Cognitive Assessment (MoCA) has been developed as one of the most popular cognitive decline evaluations and as a quick screening tool for cognitive impairment. It assesses visuospatial skills, attention, language, abstract reasoning, delayed recall, executive function, and orientation [7].

The MoCA is calculated by adding the previously mentioned non-motor symptoms [8,9]. The younger age group has recently been found to be at high risk for cognitive impairment and memory decline, despite the fact that older age groups have been shown to have an increased risk of these conditions [10]. Hely et al. [11] observed that, after ten years of observation employing a range of focused therapies, 13 out of 126 patients were still free of substantial functional constraints. In addition, 9 out of 129 patients needed help getting out of a wheelchair or bed. Due to this heterogeneity, the prognosis of future progression in PD patients is still challenging. Since cognitive impairment severely increases the burden on PD patients and their families [3], it is crucial for clinicians to predict future cognitive decline in patients with PD [1,5,12]. The MoCA has some advantages over other cognitive impairment tests, which include the following: (1) it measures global cognitive function quickly with a short administration time; (2) it involves a variety of cognitive functions; (3) it detects executive dysfunction; (4) it is sensitive to milder cognitive impairments in PD; (5) it is extensively adjusted and used in clinical care and clinical research; (6) it is used in a variety of disease conditions; and (7) it has an alternate version for different languages [1]. Since Postuma et al. [13] reported that the risk of dementia increases after 4 years, this study aimed to predict the MoCA score in year 4.

The age of onset, clinical phenotypes, and severity of PD have been shown as biomarkers to predict PD subtypes [14,15,16]. A recent study [17] classified PD subtypes based on clinical features (CF) as follows: mild, non-motor dominant, motor dominant, and severe. Recent studies [18,19] showed that age, genetic variation in Microtubule-Associated Protein Tau (MAPT), Apolipoprotein-E (APOE), gait disturbance, motor and non-motor assessments, dopamine transporter (DAT) imaging, electroencephalograms, and CSF biomarkers may contribute to the early prediction of cognitive impairment in PD. According to Mazancova et al. [20], the MoCA is the most discriminatory assessment of MCI in PD. It has been demonstrated that reduced mean caudate uptake is a biomarker of PD [21]. Additionally, reduced DAT density on DAT SPECT images has been associated with cognitive disorder, especially in the caudate [22]. In addition, decreased contralateral putamen DAT availability can be used as a longitudinal biomarker predictor of cognitive decline. According to Arnaldi et al. [17], the UPDRS-III score and DAT uptake in the caudate nucleus of the less affected hemisphere could accurately predict cognitive impairment after 4 years, primarily in de novo patients. Another study [23] discovered that lower striatum uptake, except in the posterior putamen, is linked to cognitive dysfunction in PD patients, especially frontal/executive and visuospatial function.

The machine learning (ML) method has been frequently utilized to explore hidden patterns in data by utilizing various features that may be related to the target [24,25,26,27,28]. Radiomics are textural mathematical representations that use texture to capture the spatial appearance of the tissue of interest (shape and texture) on different images [29] and provide a significant opportunity for extracting clinically relevant data from medical imaging [30,31,32]. Handcrafted radiomics features (RFs), such as intensity, morphological, and textural features, enable high-dimensional information extraction from images [33]. Moreover, a handcrafted image processing pipeline is comprised of three tightly coupled steps: feature extraction, feature selection, and ML model building. These are essential for a successful RF analysis. Small changes in each stage can have an impact on prediction accuracy and stability [34]. Furthermore, RF analysis necessitates large image data sets with the expectation that large numbers will enable researchers to overcome some of the heterogeneities inherent in clinical imaging [35]. DL, on the other hand, enhances such an image extraction workflow by automatically learning discriminant information directly from the images [25,26,34,36], so we used deep learning feathers (DF) extracted from DAT SPECT images using DL algorithms to investigate DF performance.

A recent study [37] investigated Alzheimer’s disease using an mRMR-based hybrid convolutional neural network (CNN) after extracting imaging features from MRI, SVM, and KNN classifiers were employed to classify the optimized features. Another study [38] set to predict cognitive decline using ML algorithms and CFs alone. In addition, other studies [39,40] were designed to predict cognitive decline using CFs as well as RFs and combinations of regression algorithms coupled with dimension reduction algorithms. Moreover, some studies [19,41] utilized simple linear regression to predict cognitive decline in PD. Another study [4] focused on the prediction of PD motor outcomes using CFs, RF, and combinations of prediction algorithms and dimension reduction algorithms. Some recent studies [42,43,44,45] focused on clustering PD patients via ML algorithms and CFs combined with RFs. Moreover, some studies [43,46,47] employed CFs combined with RFs and ML techniques to improve PD outcome prediction. One study [48] applied a deep learning (DL) algorithm to DAT SPECT images to predict the motor outcomes in PD. Another study [32] predicted the Levodopa amount and incremental dose via ML algorithms and CFs combined with RFs.

ML techniques improve prediction performance without the need for explicit programming effort [49]. ML approaches enable researchers to automatically build prediction pipelines by trying to capture statistically robust patterns in the analyzed data. Because the majority of prediction algorithms may not work with a large number of inputs, it is vital to select a set of relevant features to avoid overfitting and enhance prediction accuracy. To begin, we used image processing tools to segment regions of interest in DAT SPECT images. Following that, we used the standardized SERA package to extract handcrafted RFs from DAT SPECT images. Following that, we used a DL algorithm to extract DFs from the preprocessed images. We produced some datasets by combining CFs, RFs, and DFs. Finally, to optimize the prediction of the outcome, a variety of classification algorithms were chosen from various families of learner algorithms, and a feature selection algorithm was linked with classification algorithms. We investigated the prediction of the MoCA score at year 4 using clinical data and the mentioned imaging features, which were measured at baseline (year 0).

## 2. Method and Martial

In the sections that follow, we elaborate our various steps such as (i) patient data and data processing steps, (ii) ML algorithms, and (iii) analysis procedure.

### 2.1. Patient Data and Data Processing Steps

We collected 297 DAT SPECT images and their clinical data in year 0 from the Parkinson’s Progression Markers Initiative (PPMI) database (www.ppmi-info.org/data, accessed on 12 April 2022). A total of 119 and 178 patients were males and females, respectively. The average age in the baseline was 61.11 ± 9.93 years (range [32.19–84.89]). Moreover, PD participants were required to meet the following criteria at baseline: (a) have a recent idiopathic PD diagnosis; (b) not be treated for PD; (c) have a dopamine transporter deficit determined through imaging; and (d) not have dementia. Furthermore, we considered patients with a MoCA score over 26 as normal outcomes and otherwise as abnormal outcomes. Although using ML algorithms to predict cognitive decline in PD patients has the potential to be beneficial, it is essential to consider the potential ethical implications to ensure that patients’ privacy, autonomy, and well-being are protected. Therefore, PPMI, as an open-access database, has been extensively anonymized to protect patient privacy and confidentiality (www.ppmi-info.org, accessed on 12 April 2022).

#### 2.1.1. CF Collection

A total of 59 CFs included motor and non-motor symptoms such as multiple Movement Disorder Society’s Unified Parkinson’s Disease Rating Scale (MDS-UPDRS) measures, a range of task/exam performances, socioeconomic/family histories, and genetic features. These have been further listed in our publicly available resource (see data and code availability). For consistency purposes, only patients who had been off medication (such as levodopa or a dopamine agonist) for more than six hours prior to testing or imaging were included [9].

#### 2.1.2. RF Extraction

In this section, we started by considering distinct 2D slices. The properties of images are elaborated in the Appendix A. As seen in Figure 1, we utilized 12 steps to segment the dorsal striatum (DS) on DAT SPECT images and extract RFs, including (1) the use of an averaging filter to smooth 2D images; (2) the use of a contrast-limited adaptive histogram equalizations tool to enhance the contrast of the greyscale 2D images; (3) initially raising the contrast of the images; (4) using our knowledge of MRI images to cut out a region of the entire image that included the right and left striates; (5) raising the contrast of the individual pieces; (6) using 2D-order statistic filtering to store and update the grey level histogram of the picture elements in a window; (7) to enlarge the image surface, adding the results from stages 4 and 6, (8) digitalizing an image using a threshold (40 percent of maximum intensity), (9) the use of morphological closing on the binary image, (10) the use of automated image halving to separate the left and right striatum as symmetric, (11) the construction of a 3D region of interest (ROI) from 2D pictures, and (12) the use of ROIs to extract RFs. Moreover, before RF extraction, we overlaid the segmented ROIs on their SPECT images to ensure the automated segmentation procedure worked well for all images, as shown in the last image of Figure 1. Thus, we utilized our standardized SERA package [50] to extract RFs from the ROIs (left and right striatum, less affected area). We assumed each ROI (left or right striatum) in two sections, including less and more affected areas. Thus, the following segmentation procedure only enabled us to segment less affected areas for both left and right striatum, although segmenting more affected areas was difficult using this segmentation procedure. This segmentation process enabled us to automate the segmentation process. The SERA package has been extensively standardized in reference to the Image Biomarker Standardization Initiative (IBSI) [51] and studied in multi-center radiomics standardization publications by the IBSI [51] and the Quantitative Imaging Network (QIN) [52]. There were a total of 215 standardized RFs in SERA, including 29 shape features, 20 first-order (FO), 30 intensity histogram features (IH), and 136 texture features that contained co-occurrence matrices (CMs; 50 features), run-length matrices (RLMs; 32 features), size zone matrices (SZMs; 16 features), distance zone matrices (DZMs; 16 features), neighborhood grey tone difference matrices (NGTs, 5 features) and neighboring grey level dependence matrices (NGLs; 17 features).

#### 2.1.3. DF Extraction

DL algorithms have the ability to extract features automatically from data, thereby doing away with the need for manual feature engineering. In the domain of medical imaging, this is essential, since there may be several features and interactions between them that are difficult for humans to perceive. Therefore, this study aimed to use DFs extracted from autoencoder algorithms to predict the outcomes. First, a part of the images was split based on the segmentation size (22 × 28 × 40) and then applied to autoencoder algorithms for DF extraction. Figure 2 shows the structure of the 3D autoencoder algorithm [53] that was utilized to extract the DFs. In general, every autoencoder is made up of an encoder network and a decoder network. The encoding layer converts the input images to a latent representation or bottleneck, which is then converted back to the original images by the decoding layer. In an autoencoder, the number of neurons in the input and output layers must be the same. Furthermore, the training label is identical to the input data. Figure 2 depicts the encoder’s typical convolutional network architecture and hyperparameters. It includes three 3 × 3 convolutional layers, each of which is followed by a leaky rectified linear unit (LeakyReLU) and a 2 × 2 max-pooling operation. The pooling layers are employed to reduce the number of parameters. The decoder path includes three 3 × 3 convolutional layers, a LeakyReLU, and an up-sampling operation. It has three 3 × 3 convolutional layers, each of which is followed by a 2 × 2 max-pooling operation and a leaky rectified linear unit (LeakyReLU). The number of parameters is decreased by using the pooling layers. Three 3 × 3 convolutional layers, a LeakyReLU, and an up-sampling step are all included in the decoder route. For the proposed autoencoder, we were using a loss function known as binary cross-entropy. As a result, the proposed autoencoder was trained using Adam, a gradient-based optimization technique, to minimize the loss function. We extracted 5375 DFs from the bottleneck layer using DAT SPECT images and the 3D autoencoder model.

#### 2.1.4. Dataset Preparation

Figure 3 shows the data collection procedure. A total of 59 CFs that included motor and non-motor symptoms were collected, as mentioned in Section 2.1.1. As mentioned in Section 2.1.2, we also extracted 215 RFs from each ROI (left and right striatum) using the standardized SERA package. Meanwhile, we removed high correlation and low variance RFs. Moreover, we extracted 5375 DFs from the original images using a 3D autoencoder algorithm, and 153 DFs were selected with a standard deviation bigger than 0.25, as seen in Section 2.1.3. As shown in Figure 3, we combined three datasets to generate new datasets such as: (1) CFs only; (2) RFs only; (3) DFs only; (4) DFs+RFs; (5) CFs+DFs; (6) CFs+RFs; and (7) CFs+DFs+RFs.

### 2.2. ML Algorithms

#### 2.2.1. Hybrid Machine Learning Systems (HMLSs)

HMLSs refer to the combination of two or more ML techniques or algorithms to address the limitations of individual methods and improve overall performance. By combining multiple approaches, HMLSs can leverage the strengths of each method and mitigate their weaknesses. The HMLSs employed in this study include feature selection algorithms linked with different classifiers, as follows below.

##### ANOVA (Analysis of Variance) Feature Selection Algorithm

Since some studies have reported the superiority of ANOVA for feature selection tasks [54,55], this study employed it to reduce the size of features by selecting the most relevant features. Thus, the ANOVA (as elaborated in Appendix A) [56] was employed prior to the classifiers to select the most relevant features, improve prediction performance, and avoid overfitting. In addition, many studies [40,43,47] indicated that using only a fraction of the most relevant features enabled improved performance for different tasks in comparison with using all features and that most classifiers are not often able to work with much input information. Thus, it is vital to select the optimal subset of features to be utilized as inputs to avoid overfitting.

##### Classifiers

ML algorithms allow for improved task performance without being explicitly programmed [57]. Approaches based on ML aim to build classification or prediction algorithms automatically by capturing statistically robust patterns present in the analyzed data. Most predictor algorithms are not able to work with a large number of input features, and, thus, it is necessary to select the optimal few features to be used as inputs. Based on previous studies [4,32,38,39,42,43,46,47], 8 classifiers (elaborated as elaborated in Appendix A) such as AdaBoost Classifier (AdaC) [58], Bagging Classifier (BagC) [59], Gradient Boosting Classifier (GBC) [60], Random Forest Classifier (RandF) [61], Extreme Gradient Boosting Classifier (XGBC) [62], Multi-Layer Perceptron (MLP) [63], K-Nearest Neighbors Classifier (KNN) [64] and Extra Trees Classifier (ETC) [65] were experimentally selected among various families of learner algorithms. In addition, we used 5-fold cross-validation and the Bayesian optimization technique to tune the hyperparameters of the classifiers. Bayesian optimization is a powerful method that can significantly increase the performance of ML methods. Bayesian optimization is a probabilistic approach to global optimization that uses a probabilistic model to approximate the objective function and then uses this model to guide the search for the optimal solution to improve the accuracy of the predictions. In 5-fold cross-validation, data points are divided into 4 folds for training and 1 fold for testing. Moreover, 80% of training data points were utilized to train the model, and the remaining 20% were utilized to validate and select the best model. Furthermore, the remaining fold was used for hold-out testing. Meanwhile, all algorithms are elaborated in the Appendix A.

#### 2.2.2. End-to-End CNN Learning Classifier

For further investigation of MoCA prediction, we employed an end-to-end CNN algorithm. CNNs [66] are used for image-processing tasks such as image classification, object detection, segmentation, and recognition. They are designed to recognize and extract patterns from large sets of data with a grid-like structure. In image classification, CNNs are trained to classify images by learning to recognize and extract relevant features. Therefore, we first enhanced our DAT SPECT images using intensity normalization [67] and then cropped them into dimensions of (22 × 28 × 40). Then, we applied our data to a 3D-CNN, which has 13 layers and three 3D convolutional (CONV) layers made up of 32 and 64 filters, each with a kernel size of 3 × 3 × 3. Following each CONV layer is a max-pooling (MAXPOOL) layer with a stride of 2, followed by ReLU activation, and the batch normalization (BN) layer comes last. Our feature extraction block was composed primarily of three CONV-MAXPOOL-BN modules. The feature extraction block’s final output was flattened and transferred to a fully connected layer with 256 neurons. A 30% effective dropout rate was employed. The output for the binary classification problem was then comprised of a dense layer of two neurons with sigmoid activity. This was also motivated by the smaller training data points and associated memory challenges. Figure 4 shows the 3D CNN structure.

### 2.3. Analysis Procedure

We selected 297 patients who had DAT SPECT images and clinical data in year 0 from the PPMI database. A total of 59 CFs that included motor and non-motor symptoms were first collected, as mentioned in Section 2.1.1. Moreover, we segmented the ROIs using the automated procedure proposed in Section 2.1.2. Subsequently, we extracted 215 RFs from each ROI (left or right striatum) using the standardized SERA package. Furthermore, we extracted 5375 DFs from the cropped images using an autoencoder, as elaborated in Section 2.1.3. As mentioned in Section 2.1.4, we generated 7 new datasets by combining the 3 mentioned datasets (CFs, RFs, and DFs). As shown in Figure 5, we first excluded highly correlated features and then normalized the datasets using the z-score technique. ANOVA was applied to select the most relevant features among different datasets mentioned in Section 2.1.4. Subsequently, we applied combinations of the most relevant features to 8 classifiers mentioned in Section Classifiers. In addition, we cropped the images based on the segmentation size (22 × 28 × 40) and then directly applied those to a CNN, as was mentioned in Section 2.2.2. Data points were divided into 2 sections including 80% for five-fold cross-validation to select the best model based on maximum accuracy and the remaining 20% for hold-out testing to validate the best algorithms.

## 3. Results

When solely utilizing the RF dataset, as shown in Figure 6 and Figure 7, the MLP classifier followed by ANOVA, with 40 features selected, outperformed other classifiers with the highest averaged 5-fold cross-validation accuracy of 59 ± 3%. The averaged hold-out testing of 59 ± 1% confirmed our finding. Meanwhile, other classifiers such as XGBC, KNN, and AdaC reached a close performance. In the sole DF dataset, the MLP classifier with 30 features selected by ANOVA resulted in the best performance among the utilized classifiers, with the highest averaged 5-fold cross-validation accuracy of 65 ± 4% and averaged hold-out testing of 56 ± 2%, which confirmed our finding (*p*-values < 0.05 using paired *t*-test, compared to the best performance provided by the sole RF dataset). Meanwhile, RandF reached a close performance. Among the eight different classifiers applied to the CF dataset, ETC followed by ANOVA, with 20 features selected, arrived at the highest averaged 5-fold cross-validation accuracy of 77 ± 8% and averaged hold-out testing of 82 ± 2%, which confirmed the findings (*p*-values < 0.05 using paired *t*-test, compared to the best performance provided by the sole RF dataset). Moreover, ETC arrived at a close performance.

In the RF+DF, ANOVA followed by XGBC, with 30 selected features (28 DFs and 2 RFs), arrived at the best averaged five-fold cross-validation accuracy of 64 ± 7% among different HMLSs. The average hold-out testing of 59 ± 2% confirmed our finding. Meanwhile, RandF achieved a close performance. In the CF and RF dataset, ANOVA followed by ETC, with 10 selected features (10 CFs and 0 RFs), resulted in the best performance with the highest averaged 5-fold cross-validation accuracy of 78 ± 7% and averaged hold-out testing of 81 ± 2% (*p*-values < 0.05 using paired *t*-test, compared to the best performance provided by the sole RF dataset). Meanwhile, the XGBC classifier arrived at a close performance. In the CF+DF feature set, ETC followed by ANOVA, with 20 selected features (17 CFs and 3 DFs) resulted in the best performance among other classifiers with the highest averaged 5-fold cross-validation accuracy of 78 ± 9% (*p*-values < 0.05 using paired *t*-test, compared to the best performance provided by the sole RF dataset). The averaged hold-out testing of 82 ± 2% confirmed our finding.

In CF+DF+RF, ANOVA followed by ETC, with 20 selected features (15 CFs, 3 DFs, and 2 RFs) resulted in the best performance between different classifiers with the highest averaged 5-fold cross-validation accuracy of 76 ± 8% (*p*-values < 0.05 using paired *t*-test, compared to the best performance provided by the sole RF dataset). Meanwhile, an averaged hold-out testing of 83 ± 3% confirmed our finding. Moreover, RandF arrived at a close performance. The results show that the feature set including CFs followed by ETX and ANOVA outperformed other HMLSs. As a result, CFs including age, sex, baseline MoCA score, and Hopkins Verbal Learning Test (HVLT) Score played an essential role in the prediction of the outcome.

In addition, employing the original cropped DAT SPECT images directly linked with a CNN enabled an averaged 5-fold cross-validation accuracy of 69 ± 6% (*p*-values < 0.05 using paired *t*-test, compared to the best performance provided by the sole RF dataset). The averaged hold-out testing of 67 ± 2% confirmed our findings. Our finding shows that the use of HMLSs applying scaler imaging and non-imaging features outperformed the use of CNN when applied to original DAT SPECT images.

## 4. Discussion

Cognitive decline prediction in PD patients holds significant value in the management of PD and improves the quality of life [1]. Individual knowledge of cognitive progression can assist in making appropriate social and occupational decisions in relation to newly diagnosed patients’ future cognitive and physical functioning [68]. The analysis investigated in this work can help predict and detect cognitive decline in PD patients using their clinical and imaging features, and it further helps clinicians to consider treatment or prevention of PD-associated cognitive impairment. This study aimed to investigate if DFs and RFs in addition to CFs can add value to the prediction of MoCA in year 4. We proposed a couple of HMLSs, such as a feature selection algorithm followed by the classifiers, and different combinations of three datasets including CFs, DFs, and RFs. Moreover, we employed a 3D-CNN to directly predict the MoCA in year 4.

Explainable Artificial Intelligence (XAI) in the healthcare domain has the potential to improve outcomes and reduce costs, but it also raises important practical questions about how AI systems should be designed and used. This can be important, because decisions made by AI systems in healthcare can have significant consequences for patients’ lives. For example, an AI system might be used to diagnose a patient with a particular disease or to determine the appropriate course of treatment. If the AI system is not transparent and interpretable, it may be difficult for clinicians to understand how the system arrived at its decision, which could undermine trust in the system and lead to errors, though there is also evidence that too much trust in an XAI system can hamper appropriate clinical evaluations [69] so caution needs to be exercised. In any case, researchers are developing XAI solutions that are specifically tailored to the healthcare domain [70,71]. Since they are composed of many layers connected by many nonlinear interconnected interactions, neural networks are generally seen as a ‘black box’ [72]. By contrast, radiomics frameworks allow for the extraction of precise quantitative RFs (such as morphology and intensity profile) where or when included in prediction models, thereby enabling a greater level of interpretability. As a result, we attempted to use both CNN and HMLSs to use interpretable RFs. Furthermore, utilizing eight distinct classifiers allowed us to test a variety of simple (but sophisticated enough to suit a relationship between input and output well) to complex models [73]. In addition, to effectively cover the ethics of XAI, we considered sufficiently high coverage and order preservation [74]. Using multiple sizes of feature sets and varied combinations of them allowed us to cover the most available features while also studying each feature set separately.

In the first effort, when eight classifiers linked with ANOVA feature selection algorithms were applied to the different combinations of datasets mentioned above, ETC applied to the mixture of CFs and DFs feature sets enabled us to achieve the highest performance of 78 ± 9%. Moreover, ETC linked with ANOVA also resulted in a close performance of 78 ± 7% when CFs were combined with RFs. In addition, the usage of sole CFs applied to ANOVA and ETC reached a close performance of 77 ± 8%. Our findings indicated that the use of sole RFs, sole DFs, and RFs+DFs added no value to the prediction task. As shown in the result section, the majority of the relevant features selected by ANOVA belonged to CFs. As a result, CFs such as age, sex, baseline MoCA score, and Hopkins Verbal Learning Test (HVLT) score played a substantial role and added significant value to the prediction task at year 4. Furthermore, the results indicated that ANOVA linked with ETC could significantly enhance prediction performance. In addition to the main part of the study, we directly applied the cropped original DAT SPECT images to a 3D-CNN to predict the MoCA in year 4. Therefore, the use of a 3D-CNN with a performance of 69 ± 6% added no value to the outcome prediction compared to the use of the imaging feature combined with CFs and the optimized HMLSs.

Similarly, in our previous study [40], we achieved a minimum MAE of 1.68 ± 0.12 by using an HMLS. This included the Non-dominated Sorting Genetic Algorithm (as a feature selection algorithm) and the Local Linear Model Trees used as a classifier on sole CFs in year 0. In this study, using feature selection algorithms significantly added value to the prediction task. Moreover, no imaging features were employed. Previous research [32,40,43] found that imaging features, in addition to the use of CFs, have the potential to enhance PD outcome prediction. For instance, employing dopamine transporter (DAT) SPECT imaging to identify individuals who are symptomatic without evidence of dopamine deficiency (SWEDD), has already had an impact on patient recruitment in clinical studies [75,76]. Furthermore, there is the opportunity for additional imaging data analysis in addition to the use of imaging features such as RFs and DFs [77].

Neuroimaging is frequently used to diagnose and predict neurodegenerative brain diseases such as Alzheimer’s disease (AD) and PD. Researchers attempted to classify magnetic resonance imaging (MRI) images with ResNet18 and DenseNet201 in a study [78], and the proposed model achieved 98.86% accuracy, thus indicating that advanced DL with MRI images can be used to classify and predict neuro-degenerative brain diseases such as Alzheimer’s disease. In addition, neuroimaging has been employed in the diagnosis of different medical disorders. In a work by Ke et al. [79], they proposed using the adaptive independent subspace analysis method to uncover relevant electroencephalogram activity in MRI scan data. The classification results were obtained using the K-Nearest Neighbor classifier with 10-fold cross-validation, which achieved 94.7% accuracy, and they showed that the image subspace texture-based classification can be utilized for assessing differences between normal and autistic brain fMRI image slices within brain tissue. Moreover, Yousaf et al. [80] indicated that a mixture of biological and neuroimaging markers improved the prediction of cognitive decline. Furthermore, Leung et al. [81] indicated that the use of imaging features significantly improves the prediction of cognitive impairment in PD. It is also feasible to use fusion methods to diagnose and predict neurodegenerative disorders. In another study, Odusami et al. [82] suggested a multimodal fusion-based strategy that used a discrete wavelet transform (DWT) and VGG16 to detect AD. The model’s high accuracy demonstrated that the proposed approach is useful in incorporating information from several neuroimaging modalities and using it to effectively diagnose AD.

Our previous study [77] focused on motor outcome prediction in PD, thus demonstrating that RFs, when combined with CFs, can significantly improve outcome prediction. We also demonstrated that combining feature selection algorithms followed by prediction algorithms resulted in very accurate motor outcomes [4] and cognitive outcome prediction [38]. Overall, this study shows the importance of using optimized HMLSs and CFs, in addition to imaging features such as RFs and DFs, to significantly enhance cognitive decline prediction in PD patients. Since clinical outcome prediction is more challenging because PD is a progressive neurological and heterogeneous disease, predicting disease outcomes in this study can help to improve patient care (such as diagnosis and treatment planning), resource allocation, and research efforts, which can empower patients to take a more active role in their care.

Selection bias and measurement bias are two potential biases that may influence the results of a study or analysis using ML algorithms. Selection bias can occur when the data used to train or test a ML algorithm is not representative of the population or phenomenon of interest. Moreover, measurement bias can occur when the data employed in the study is not accurately or consistently measured. Thus, we tried to improve selection bias by selecting the maximum number of patients who had images and clinical data, although the limited size of a dataset is still a limiting factor in the outcome prediction. As such, to maximize our numbers, we had to select a set of 297 patients for which images and clinical data were available to all patients. Moreover, one collaborative physician was invited to double-check segmentations delineated from DAT SPECT. Our clinical data collection, DF, and RF extraction procedure were also double-checked to improve measurement bias. Moreover, ML algorithms still have limitations related to overfitting, underfitting, interpretability, and fairness, which have been improved through the employment of optimization techniques, relevant features selection, maximum data point selection, and suitable algorithm selection from our previous studies.

In our work, we used a feature selection algorithm to reduce the number of features (for size reduction) to avoid overfitting, although it was possible to utilize extraction algorithms such as PCA, which we hope to explore in future work. At the same time, we believe feature selection might be more clinically informative than feature extraction, thus providing insights as to which features are most important, whereas, in feature extraction, features are combined and transformed into new dimensions and, thus, may not be easily interpretable. Since the study’s sample size is limited, more research with larger datasets is required to generalize our findings.

## 5. Conclusions

Our research found that using appropriate HMLSs with both CFs and imaging features allowed for accurate prediction of MoCA in year 4. Specifically, ANOVA and ETC applied to datasets including CFs enabled us to achieve an average accuracy of 78 ± 9%. The majority (though not all) of relevant features selected by ANOVA were CFs, e.g., age, sex, baseline MoCA score, and Hopkins Verbal Learning Test (HVLT) score, which played vital roles in the prediction of MoCA at year 4 in PD. Furthermore, the use of a 3D-CNN with a performance of 69 ± 6% added nothing to the prediction of the result compared to the use of imaging and clinical characteristics using improved HMLSs. Overall, the findings show that baseline CFs play an important role in predicting likely future cognitive decline (or the MoCA including visuospatial skills, attention, language, abstract reasoning, delayed recall, executive function, and orientation), and the imaging features surprisingly added no value to the prediction of cognitive decline in year 4. This leads to the conclusion that CFs acquired at the initial stage of the PD, such as age, gender, baseline MoCA score, and Hopkins Verbal Learning Test (HVLT) score, can help clinicians predict future cognitive decline using HMLS including ANOVA and ETC, thus providing a better prevention and management plan.

## Figures and Tables

**Figure 1 diagnostics-13-01691-f001:**
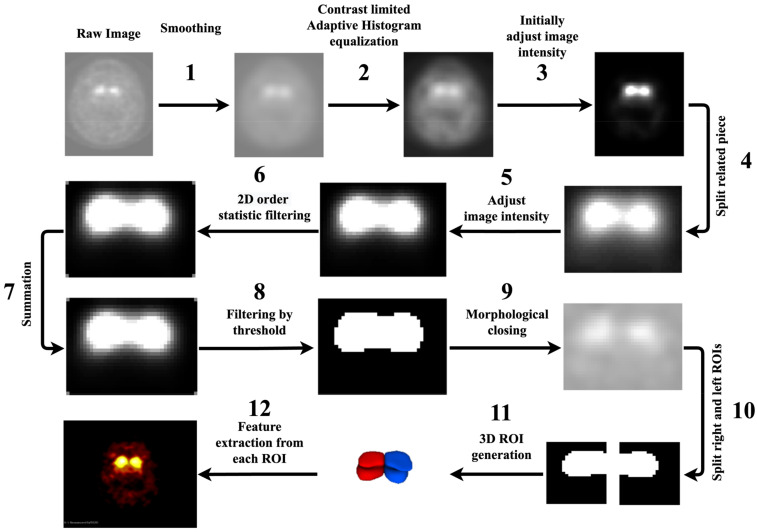
Segmentation process based on SPECT through 12 above-mentioned steps. Image 1 shows raw DAT SPECT. Images 2 to 11 show the result of the image processing steps. Image 12 shows a 3D segmentation of the Dorsal Striatum, and image 13 shows a fusion of DAT SPECT and the segmentations.

**Figure 2 diagnostics-13-01691-f002:**
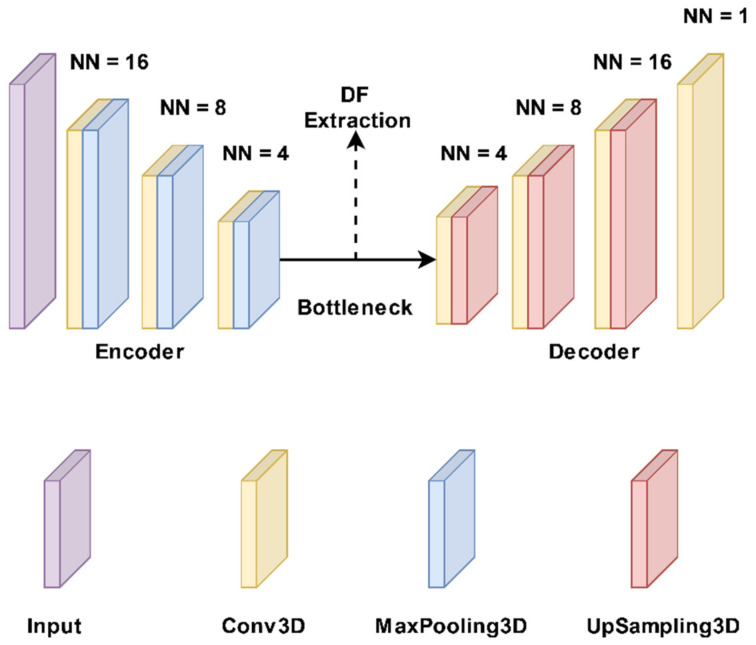
The autoencoder model structure. It has three 3 × 3 convolutional layers with each being followed by a 2 × 2 max-pooling operation and a leaky rectified linear unit (LeakyReLU). Three 3 × 3 convolutional layers, a LeakyReLU, and an up-sampling operation make up the decoder route. NN: Neuronal number.

**Figure 3 diagnostics-13-01691-f003:**
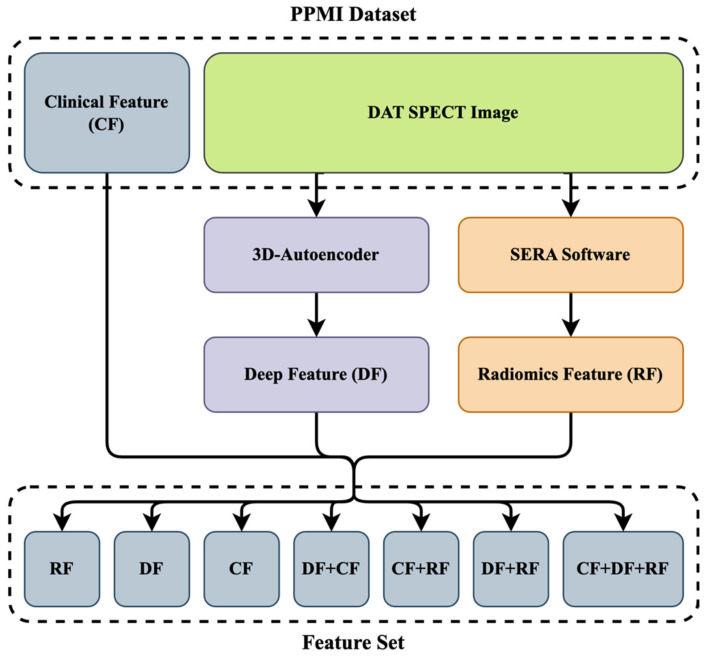
The diagram of the feature set was used to predict the MoCA score in year 4. After extracting clinical and DAT SPECT images, we extracted DFs and RFs from the original images using a 3D autoencoder algorithm and standardized SERA software. We finally combined these datasets with CFs to generate 8 new feature sets. Deep feature: DF. Radiomics feature: RF. Clinical feature: CF.

**Figure 4 diagnostics-13-01691-f004:**
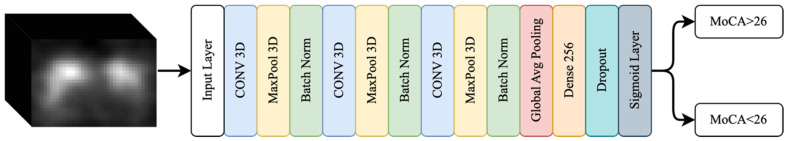
A schematic diagram of the proposed convolutional neural network consisting of 13 layers including three 3D convolutional (CONV) layers, three max-pooling (MAXPOOL) layers, and three batch normalization (BN) layers passed to a fully connected layer with 256 neurons.

**Figure 5 diagnostics-13-01691-f005:**
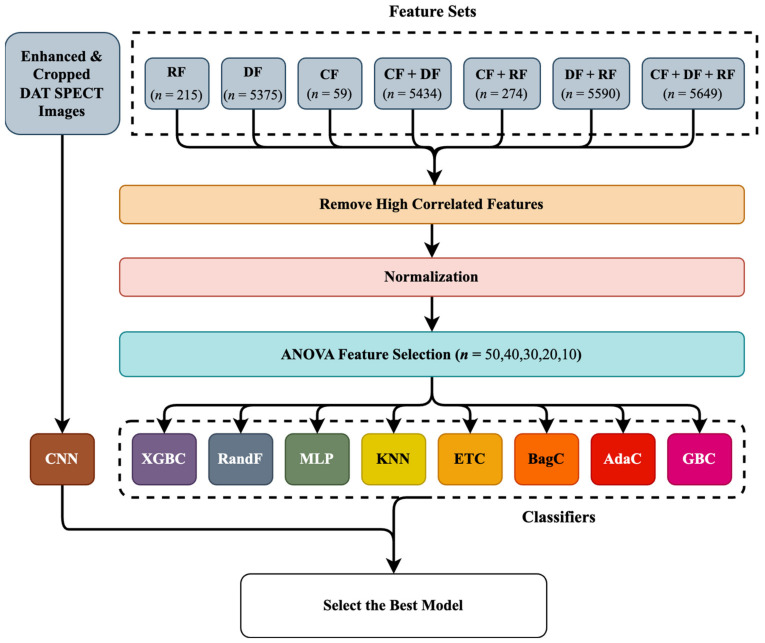
Diagram of the study procedure. After making 7 feature sets, the highly correlated features were removed, and the remaining features were normalized. Subsequently, ANOVA was applied prior to 8 classifiers to select the relevant features. Moreover, we also applied the preprocessed images to a convolutional neural network (CNN). *n*: number of features. AdaBoost Classifier: AdaC. Bagging Classifier: BagC. Gradient Boosting Classifier: GBC. Random Forest Classifier: RandF. Extreme Gradient Boosting Classifier: XGBC. Multi-Layer Perceptron: MLP. K-Nearest Neighbors: KNN. Extra Trees Classifier: ET. Deep feature: DF. Radiomics feature: RF. Clinical feature: CF.

**Figure 6 diagnostics-13-01691-f006:**
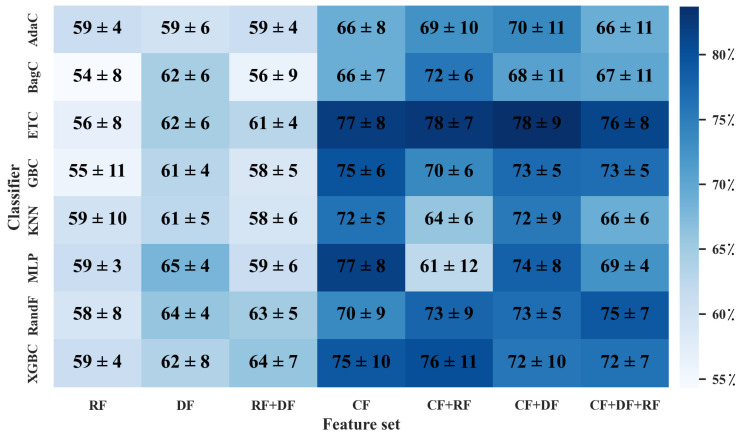
Heatmap of averaged accuracy and standard deviation (STD) for MoCA prediction in five-fold cross-validation. All values are reported as a percentage. X axis indicates different combination of datasets (CFs, DFs, RFs), and Y axis indicates classifiers mentioned in Section Classifiers. CF: Clinical features. DF: deep features. RF: radiomics features.

**Figure 7 diagnostics-13-01691-f007:**
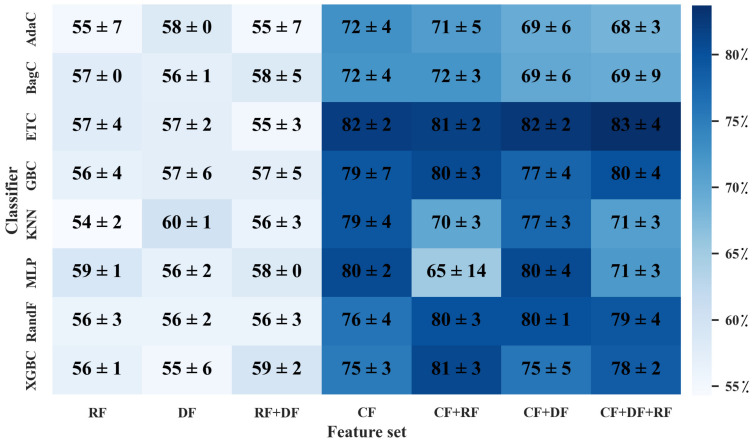
Heatmap of averaged accuracy and standard deviation (STD) for MoCA prediction in hold-out testing. All values are reported as a percentage. X axis indicates different combination of datasets (CFs, DFs, RFs), and Y axis indicates classifiers mentioned in Section Classifiers. CF: Clinical features. DF: deep features. RF: radiomics features.

## Data Availability

All datasets were collaboratively pre-processed by Qurit Lab (Quantitative Radiomolecular Imaging and Therapy, qurit.ca) and the Technological Virtual Collaboration Corporation Company (TECVICO Crop., tecvico.com). All codes were also developed collaboratively. All codes (including predictor algorithms and feature selection/extraction algorithms) and datasets are publicly shared at: https://github.com/Tecvico/Prediction_Cognitive_Decline_in_Parkinson-s_Disease_using_Deep_and_Handcrafted_RF.

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
