# Peer review of "Prediction of Cognitive Decline in Parkinson’s Disease Using Clinical and DAT SPECT Imaging Features, and Hybrid Machine Learning Systems"

_diagnostics, 2023, doi:10.3390/diagnostics13101691_

Round 1

Reviewer 1 Report

The authors aimed to estimate the scores of Montreal Cognitive Evaluation (MOCA) in Parkinson's disease patients in the 4th year using handmade radiomics (RF), deep (DF) and clinical (CF) properties applied to hybrid machine learning systems (base line). For this, different classifications accepted in the literature were used in the study. Similar studies on the subject should be included. At the end of the Introduction section, a paragraph for the organization of the article should be added. Sound data were mostly used to detect Parkinson's disease. The authors should present this way to the literature and innovation. Different success measurement parameters are used to evaluate the results. I recommend you to review the relevant work (https://doi.org/10.1002/ima.22632). Autoencoder structure is given in Figure 3. Explaining Figure 3 is important. The initial feature numbers in Figure 5 and the number of features after the selection process should be specified.

Author Response

Thank you for your valuable time to review our work. We remain open to further feedback. We now addressed your valuable comments. Please find the document attached.

Reviewer 2 Report

I have several concerns about the paper "Prediction of Cognitive Decline in Parkinson's Disease using Clinical, DAT SPECT Imaging Features, and Hybrid Machine Learning Systems." Firstly, the paper lacks clarity in its presentation of the research methodology. The authors do not provide a clear description of the data collection process, the sample size, or the criteria used to select the participants. This lack of transparency makes it difficult to assess the validity of the study's findings. Secondly, the paper does not adequately address the limitations of the study. The authors do not discuss the potential biases that may have influenced the results, such as selection bias or measurement bias. Additionally, the authors do not provide a clear explanation of the limitations of the machine learning algorithms used in the study. This lack of discussion undermines the credibility of the study's conclusions. Thirdly, the paper does not provide a clear explanation of the clinical implications of the study's findings. The authors do not discuss how the results of the study could be used to improve the diagnosis or treatment of Parkinson's disease. This lack of discussion makes it difficult to understand the practical significance of the study's findings. Overall, I believe that the paper "Prediction of Cognitive Decline in Parkinson's Disease using Clinical, DAT SPECT Imaging Features, and Hybrid Machine Learning Systems" is not suitable for publication in its current form. The paper lacks clarity in its presentation of the research methodology, does not adequately address the limitations of the study, and does not provide a clear explanation of the clinical implications of the study's findings.

I suggest the following improvements of this manuscript for the authors:

1. Provide a clear and detailed description of the data collection process, sample size, and participant selection criteria to increase transparency and improve the validity of the study's findings.

2. Address the potential biases that may have influenced the results, such as selection bias or measurement bias, and discuss the limitations of the machine learning algorithms used in the study.

3. Clearly explain the clinical implications of the study's findings and how they could be used to improve the diagnosis or treatment of Parkinson's disease.

4. Consider using a larger dataset to increase the accuracy and generalizability of the study's findings.

5. Provide a more detailed explanation of the methodology used to segment the imaging data and how it was used in the analysis.

 6. Consider using a more diverse set of imaging features to further improve the accuracy of the predictions.

7. Discuss the potential ethical implications of using machine learning algorithms to predict cognitive decline in Parkinson's disease patients.

8. Provide a more detailed explanation of the HMLSs used in the study and how they were optimized to improve the accuracy of the predictions.

9. Consider using a more diverse set of classifiers to further improve the accuracy of the predictions.

10. Provide a more detailed discussion of the limitations of the study and how they may impact the interpretation of the results.

Author Response

Thank you for your valuable time to review our work. We remain open to further feedback. We now addressed your valuable comments carefully. Please find the document attached.

Reviewer 3 Report

I am really grateful to review this manuscript. In my opinion, this manuscript can be published once some revision is done successfully. This study used 9 machine learning models and various combinations of clinical, deep and radiomic features from 297 patients to achieve the accuracy of 83% for the prediction of the Montreal Cognitive Assessment Score. I would argue that this is a rare achievement. However, the issue of explainable artificial intelligence requests due attention now hence I would like to ask the authors to address this issue in Discussion. 

Author Response

Thank you for your valuable time to review our work. We remain open to further feedback. We now addressed your valuable comment. Please find the document attached.

Round 2

Reviewer 1 Report

The authors have corrected the desired deficiencies in the revision. However, the writing formats of the sources are different from each other. The references should be arranged according to the format of the journal.

Author Response

Response: Thank you for your valuable time to review our work. We now correct the remaining deficiencies and make consistent all references throughout the manuscript. We remain open to further feedback.

Comment: The authors have corrected the desired deficiencies in the revision. However, the writing formats of the sources are different from each other. The references should be arranged according to the format of the journal.

Response: Thank you for your valuable time to review our work. We now correct the remaining deficiencies and make consistent all references throughout the manuscript. We remain open to further feedback.

Reviewer 2 Report

Dear authors, thank you for submitting the revised version of your research paper "Prediction of Cognitive Decline in Parkinson's Disease using Clinical, DAT SPECT Imaging Features, and Hybrid Machine Learning Systems" to Diagnostics journal. While your paper presents an interesting approach to predicting cognitive decline in Parkinson's disease and the revision has improved the quality of the paper, there are several areas that still need improvement before it can be accepted for publication. Below are some suggestions for the 2nd revision:

1. The methodology section needs to be more detailed and clear. Please provide a step-by-step explanation of the algorithms and techniques used, and why they were chosen over other options. This will help readers understand the exact steps taken to arrive at the results.

2. The paper lacks a clear research question or hypothesis. Please define a specific aspect of cognitive decline that you are trying to predict and explain why this is important. This will help readers evaluate the significance of the results.

3. Discussion of the related studies needs to be improved by providing better contextualisation within the neuroimaging research field. The authors are encouraged to check “Adaptive independent subspace analysis of brain magnetic resonance imaging data”, “An intelligent system for early recognition of Alzheimer’s disease using neuroimaging”, and “Pixel-level fusion approach with vision transformer for early detection of Alzheimer’s disease”.

4. The paper still lacks a clear conclusion or summary of the findings. Please provide a clear explanation of what the results mean for the field of Parkinson's disease research or for clinical practice.

5. The writing in the paper still needs improvement. Please ensure that the paper is well-organized, concise, and free of grammatical errors.

6. The paper briefly mentions the issue of Explainable Artificial Intelligence (XAI) in healthcare, but does not provide a clear explanation of how the study addresses ethical and practical concerns. Please provide a more detailed discussion of how the study addresses these concerns.

I hope that these suggestions will be helpful in revising your paper. I look forward to receiving your revised manuscript.

Author Response

Thank you for your valuable time to review our work. We have now addressed your concerns step-by-step, as attached. We remain open to further feedback.

Round 3

Reviewer 2 Report

The authors have revised well. The manuscript can be accepted for publication.